# Human genetic evidence enriched for side effects of approved drugs

Eric Vallabh Minikel [1], Matthew R. Nelson [2,3]*

1 Stanley Center for Psychiatric Research, Broad Institute, Cambridge, Massachusetts, United States of America, 2 Deerfield Management Company, L.P, New York City, New York, United States of America, 3 Genscience LLC, New York City, New York, United States of America

* mnelson@genscience.com

## Abstract

Safety failures are an important factor in low drug development success rates. Human genetic evidence can select drug targets causal in disease and enrich for successful programs. Here, we sought to determine whether human genetic evidence can also enrich for labeled side effects (SEs) of approved drugs. We combined the SIDER database of SEs with human genetic evidence from genome-wide association studies, Mendelian disease, and somatic mutations. SEs were 2.0 times more likely to occur for drugs whose target possessed human genetic evidence for a trait similar to the SE. Enrichment was highest when the trait and SE were most similar to each other, and was robust to removing drugs where the approved indication was also similar to the SE. The enrichment of genetic evidence was greatest for SEs that were more drug specific, affected more people, and were more severe. There was significant heterogeneity among disease areas the SEs mapped to, with the highest positive predictive value for cardiovascular SEs. This supports the integration of human genetic evidence early in the drug discovery process to identify potential SE risks to be monitored or mitigated in the course of drug development.

**Data availability statement:** An analytical dataset and source code are available at https://github.com/ericminikel/genetics_side_effects/

## Author summary

Side effects are a major contributor to the high failure rates of clinical drug development. There are numerous anecdotes where a genetic association between a human gene and a trait mirrors the association between a drug targeting the protein product of that gene and a side effect similar to the trait. These anecdotes have generated interest in using human genetics to predict drug side effects, but to date there are very few systematic studies examining the predictive value in this approach. Here we combine human genetic datasets spanning common and rare diseases with a database of side effects that are noted on approved drug labels. We find that overall side effects are twice as likely to occur if they are similar to a trait where the drug target gene has a genetic association. This suggests that human genetic evidence can be useful in identifying potential on-target safety liabilities.

and are sufficient to reproduce all figures and statistics herein.

**Funding:** This work was supported by Deerfield Management Company, L.P. and Genscience, LLC in the form of salaries to MRN and EVM. The funders had no role in study design, data collection and analysis, decision to publish, or preparation of the manuscript.

**Competing interests:** I have read the journal's policy and the authors of this manuscript have the following competing interests:MRN is an employee of Genscience and Deerfield. EVM is a consultant to Deerfield Management Company, L.P. and Genscience, LLC. EVM also acknowledges research support, unrelated to the current work, from Ionis, Gate, Sangamo, and Eli Lilly, and consulting fees, unrelated to the current work, from Eli Lilly and Alnylam.

## Introduction

Safety issues are a major contributor to drug candidate failure, with clinical safety findings accounting for 25% of drug program terminations in Phase I-II [1]. The causal evidence of human genetics between drug targets and phenotypic outcomes can provide insights into potential on-target safety liabilities of drug candidates before development has even begun [2]. There are many anecdotes of adverse events predicted by genetics because they are similar to traits genetically associated with a drug target [2]. Conversely, there are also examples of drug tolerability being supported by the lack of negative consequences of genetic loss of function in the drug target gene in humans [3]. Inspired by these anecdotes, phenome-wide association studies [4], curation of loss-of-function variants [5], and recall-by-genotype of rare homozygous loss-of-function participants [6] have been used to evaluate potential on-target liabilities.

Given the strong evidence that genetic evidence supporting the connection between target and indication for a drug increases the probability of clinical success, presumably by predicting on-target pharmacology, it is to be expected that genetic evidence will also predict undesired on-target pharmacology, namely, side effects (SEs). Nevertheless, to date, there is limited systematic support for the predictive value of genetic evidence for SEs. One reason is that SEs can be caused by on or off target biology. On-target SEs occur as a result of engaging the intended drug target. Off-target SEs occur as the result of unspecific drug protein binding, impacting biological pathways unrelated to the intended therapeutic target. Still other SEs are simply adverse events that get reported and may become enshrined in drug labels despite not being causally related to the drug at all: some may be coincidental, or may be associated with the indication for which the drug is being prescribed in the first place. The study of how to use genetics to anticipate potential SEs thus suffers from dilution of on-target liabilities by off-target and non-associated SEs. It has been shown that drugs with SEs are more likely to bind off-target proteins associated with Mendelian diseases similar to the SE [7], but this approach does not assess potential on-target effects at the time of target selection early in the drug discovery process. One systematic study of clinical trial SEs has shown a 1.80-fold enrichment for SEs affecting the organ system in which the drug target has genetically associated traits [8], which dropped to 1.55-fold when drugs with pharmacologic action in that same organ system were removed. To date, no study has systematically examined the enrichment of SEs for genetic associations at a finer grain of SE-trait similarity. Moreover, no study has examined the positive predictive value of human genetic evidence for SE prediction.

Recently, in re-assessing the utility of human genetics for predicting drug approval [9], we mapped human genetic evidence and drug indications to the Medical Subject Headings (MeSH) ontology and constructed a similarity matrix among MeSH terms using Lin-Resnik similarity, which is based on both term co-occurrence and position in the ontological hierarchy [10,11]. This approach permitted us to examine the enrichment of successful drug target-indication pairs for genetically associated gene-trait pairs where the target is the same and the indication and trait are similar. Here, we adapt this approach to SEs, joining reported drug-SE pairs to genetically supported gene-trait pairs, and estimating the enrichment above background among the set of all drug-SE pairs. We demonstrate that SEs reported in approved drug labels are enriched for genetic evidence. By assigning quantitative similarity scores, we are able to test the sensitivity of such enrichment to the similarity threshold, to the source of genetic evidence, and to potential confounders. Finally, we estimate the positive predictive value of human genetic evidence, and examine the how SE frequency, specificity, and severity impact the value of genetically-informed predictions across different types of SEs.

## Results

Given the paucity of systematic data about statistically enriched SEs observed in clinical trials, we chose to focus on SEs captured in approved drug labels and package inserts in Side Effect Resource (SIDER) [12,13], to which we joined a database of human genetic evidence [14]. This resulted in 2,094 unique SEs (MeSH terms), and 567 unique drugs with at least one human target, one SE and one approved indication, or 1,187,298 possible drug-SE pairs (Tables A-D in S1 Table). Of these possible pairs, SEs were observed (reported in the drug label) for 64,481, yielding an overall base rate (marginal probability of an SE being observed for a given drug) of 5.4%. Throughout, our analysis will use the universe of possible drug-SE pairs, and we will use the term "observed" to refer to those drug-SE pairs where the SE occurs in the drug label.

The primary analysis of interest is the relationship between drug-SE pairs and the presence of genetic evidence between the gene encoding the drug target and a trait similar to the SE. For instance, variants in *SCN5A* are associated with cardiac arrhythmia, and topiramate, which is indicated for migraine, targets *SCN5A* and has cardiac arrhythmia listed as a SE. When we defined genetic evidence as SEs and traits with ≥0.9 similarity, we found genetic evidence to be strongly enriched among observed SEs (OR = 2.3, 95% CI = 2.2-2.5, P = 7.1e-93, Fisher's exact test).

We explored several possible confounders that could affect this observed enrichment (S1 Fig and Table E in S1 Table). One such confounder would be if SEs similar to genetically studied traits occur more often. We therefore restricted our analysis to drug-SE pairs where the SE has been studied genetically (see "Genetic insight" in Methods). For example, 234 drugs had chills listed as a SE, but our database did not contain any genetic associations for chills; thus, this SE was excluded from analysis. This filter left 45,474 observed drug-SE pairs. Another confounder would be if SEs that are not causally related to the drug, but simply co-occur with the drug's indication, become enriched because they are similar to the indication, which in turn is genetically associated to the target. We therefore also removed drug-SE pairs where the drug is approved for an indication with a similarity ≥0.9 to the SE. For example, *ADRB1* is genetically associated to cardiac arrhythmia, but bisoprolol, which targets *ADRB1* and is both indicated for cardiac arrhythmia and also has cardiac arrhythmia listed as a SE, was removed. These changes had a modest effect, reducing the OR to 2.0 (95% CI = 1.8-2.1, P = 2.6e-58) in combination (S1 Fig and Table F in S1 Table). We retained both filters for all subsequent analyses.

Given the strong prior evidence that human genetics can predict on-target pharmacology, we expected to see ORs greater than 1, and therefore to reject the null hypothesis that genetic evidence is not enriched among observed SEs. In this study, we were particularly interested in the characteristics of the SEs and genetic evidence that were enriched, and the relationship between the SEs and genetic evidence or approved indications to aid in using genetic evidence to make better target selection and safety risk decisions.

We further investigated the relationship between the SE and the traits with genetic evidence by examining the sensitivity of the OR to the SE-trait similarity threshold. We observed ORs at least slightly above 1 across all thresholds tested (minimum OR = 1.14). We wondered whether this might reflect that more pleiotropic targets, which have more associations, also have more SEs. However, there was no correlation between the count of observed drug SEs an the count of unique traits to which its target is genetically associated (rho = -0.01, P = 0.72, Pearson's correlation). Instead, the slight enrichment observed even at low similarity thresholds might simply recapitulate the previously reported enrichment at the level of organ system [8] (Fig 1A and Table G in S1 Table), perhaps by selecting for tissues in which the drug is present and the target expressed. Regardless, the OR inflected at a similarity threshold of 0.75 and rose sharply

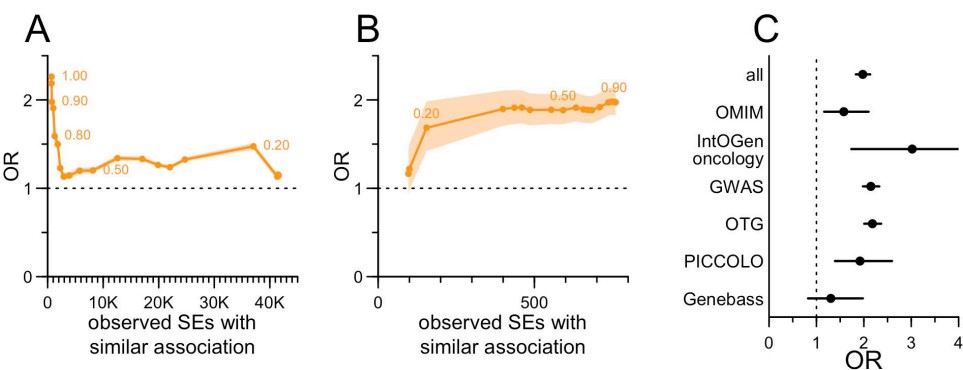

**Fig 1. Predictive value of human genetic evidence for labeled drug side effects.** A) Sensitivity of OR to the threshold for similarity of SE to genetically studied traits. Selected similarity thresholds are annotated. The resulting number of drug-SE pairs with similar genetic evidence is shown on the x axis and OR is shown on the y axis. Threshold for removal of SEs with similar indications is fixed at 0.9. B) As in A, but here the threshold for SE-trait similarity is fixed at 0.9 while the similarity threshold between the SE and approved indications is varied. C) OR for several sources of genetic evidence contributing to this study.

thereafter, reaching OR=2.0 at a threshold of similarity 0.90. Thus, genetic evidence is most enriched among SEs that are extremely similar to the genetically associated trait.

Varying the threshold for removal of SEs based on their similarity to approved indications had minimal impact (Fig 1B and Table H in S1 Table). For instance, removing all SEs with similarity ≥0.9 to the indication yielded 755 SEs with genetic evidence, and an OR of 2.0, while removing all SEs with similarity ≥0.5 to the indication yielded 592 such SEs and an OR of 1.9. The OR was substantially impacted only when we removed all SEs with similarity ≥0.15 to the indication, a threshold where the vast majority of the data were removed. At the 0.9 similarity threshold selected here, only 334/45,475 (0.73%) observed drug-SE pairs were removed from analysis due to the similar indication filter.

The sensitivity testing results support the ≥0.9 similarity thresholds used for both metrics. We next examined whether the source of genetic evidence had any influence on this enrichment (Fig 1C and Table I in S1 Table), and found little difference, though germline oncology evidence had the highest levels of enrichment (S2 Fig and Tables J-K in S1 Table).

We next investigated how the enrichment of genetic evidence for SEs interacts with genetic evidence supporting the drug indication, termed genetic support. Because each drug may have many approved indications, we focused on the indications most similar to the reported SEs. Using our previous dataset, we found genetic support for 287/1,993 (14.4%) of drug-target-indication tuples in our database, comprising 20.1% of all possible drug-SE pairs. The base rate of SEs was similar among drugs with and without genetic support: 4.9% for genetically supported drugs and 5.6% for unsupported drugs, suggesting that genetically supported drugs are no more or less likely to display SEs than other drugs. When no filter against SEs similar to indications is applied, drug-SE pairs with genetic evidence for the SE and with genetic evidence for the indication overlap considerably more than expected by chance (OR = 2.5, P = 2.5e-34). Naturally, because these are by definition instances where the SE, the indication, and the genetic association are all highly similar to one another, many are removed by the similar indication filter that we used in the analyses throughout this study. Removing drug-SEs where the SE and indication share a similarity ≥0.9 results in a slightly lower enrichment that is similar both for drug-indication pairs with (OR = 2.1, P = 2.3e-20), and without genetic support (OR = 1.9, P = 3.1e-41). Given this, we conclude that genetic support for the drug indication is not a major confounder for any of the SE-related analyses in this study.

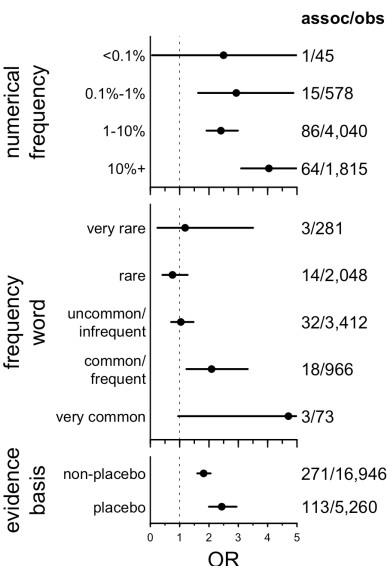

**Fig 2. Impact of SE modifiers on genetic evidence enrichment.** *Because frequency and evidence basis are only defined for observed SEs, the OR indicated here is the enrichment of genetic evidence conditioned on an SE being observed with the indicated modifier. The assoc/obs fraction indicates in the denominator the number of drug-SE combinations observed with the indicated properties and in the numerator the number of those that have genetic evidence. The ordering of the frequency words is based on ref. [15].*

SIDER provides several SE modifiers extracted from the text, including SE frequency among patients as numerical estimates or descriptive terms and whether the SEs were supported by placebo or non-placebo-controlled studies. Where numerical frequencies were available, we found that genetic evidence enrichment increases with increasing SE frequency (P = 7.4e-8, binomial logit, Fig 2 and Tables L-M in S1 Table). The same was true for descriptive frequency terms ranked by their reported perceived numeric values [15] (P = 0.061 for the linear term in a binomial logit orP = 0.0018 when analyzed using term ranks; Fig 2 and Tables N-P in S1 Table). Estimates of enrichment were higher for SEs backed by placebo-based evidence, but that difference was not statistically significant (P = 0.079, binomial logit, Fig 2 and Tables Q-R in S1 Table).

We further explored enrichment of genetic evidence based on the number of different drugs for which an SE was reported, without respect to the underlying target gene. Some SEs are highly drug-specific, for example, bezoar is a labeled SE for only 2 drugs (lansoprazole and nifedipine), while other SEs are reported for a huge number of drugs, for instance, nausea and headache are each labeled for >500 drugs. We found that the enrichment for genetic evidence was strongest for SEs observed for greater than one, but fewer than ten drugs, and decreased as the number of drugs increased (Fig 3A and Table S in S1 Table). This poses a challenge for practical utility of human genetics in predicting SEs. The SEs that are most informed by genetic evidence are more drug-specific, and highly drug-specific SEs necessarily have a low base rate, resulting in relatively low predictive values (PPVs; probability of observing an SE given genetic evidence, Fig 3B and Table S in S1 Table).

We also considered crowdsourced severity rankings of the observed SEs [16], grouping these by quartile. For instance, coma and death both rank in the top quartile of severity, while euphoria and tooth discoloration are in the bottom quartile. We found that OR was also positively, though non-monotonically, associated with SE severity (P = 3.0e-23, binomial logit; Fig 3C and Tables T-U in S1 Table), while the base rate was slightly lower for the more severe

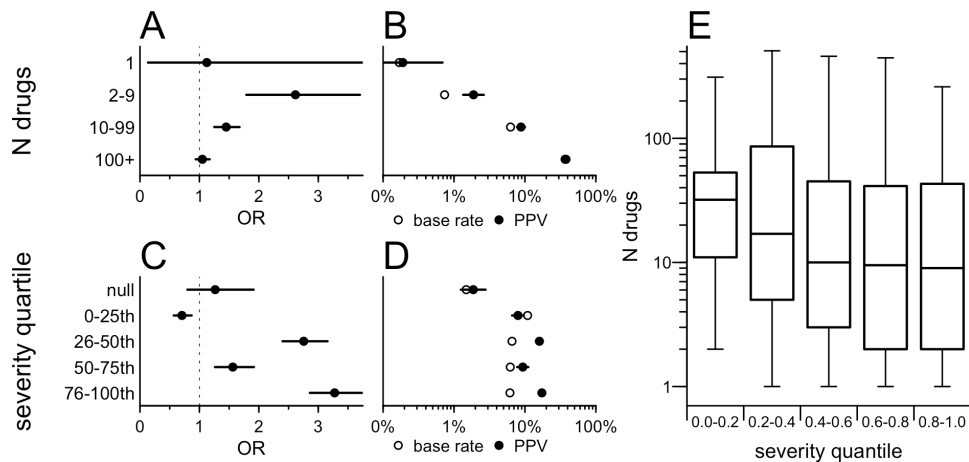

**Fig 3. Relationship between SE specificity and severity, and the predictive value of genetic evidence.** A) OR for enrichment of genetic evidence, binned by the number of drugs for which SE was observed. The unit of analysis is drug-SE pairs; thus, the number of observed drug-SE pairs is necessarily higher for those SEs observed for a larger number of drugs, hence the tighter confidence intervals in the "100+" bin compared to the "1" bin. B) Base rate (proportion of drugs reporting the SE) and positive predictive value (proportion of drugs with genetic evidence for the SE) binned as in A. Note that the higher base rate for those SEs observed for a larger number of drugs is tautological. C) OR by quartiles of SE severity. D) Base rate and positive predictive value binned as in C. E) Boxplot of specificity by severity bin; x axis is quintiles of severity, y axis is number of drugs for which the SE is observed. Boxes represent means and interquartile ranges. Whiskers represent minimum and maximum datapoints that fall within 1.5 interquartile ranges of the box.

SEs (Fig 3D and Table T in S1 Table). SE severity and specificity were themselves correlated (P = 2.1e-4, linear regression, log(n_drugs) ~ severity; Tables V-W in S1 Table) with more severe SEs tending to be observed for fewer drugs (Fig 3E). In other words, the base rate was lowest for the most severe quartile of SEs (6.1%, versus 10.9% for the least severe quartile of drugs). Drugs without a severity score assigned had the lowest base rate of all, perhaps simply reflecting that less commonly encountered SEs were less likely to be included in the original survey-based study used to rank severity.

To better understand the value of genetic evidence on SE risk, we next binned SEs by top level disease headings of the MeSH ontology (Fig 4 and Table X in S1 Table), revealing substantial effect size heterogeneity (P <1e-15, CMH test; Fig 4A) and in the potential utility of genetic evidence (Fig 4B). The endocrine category, including SEs such as diabetes mellitus and hypothyroidism, had the largest effect (OR = 6.5) with a low base rate (1.9%) and a PPV of 10.5%, and were moderately severe (Fig 4C and Table X in S1 Table). In contrast, cardiovascular, including SEs such as bundle-branch block, sick sinus syndrome, and long QT syndrome, had a combination of both high base rate and high OR resulting in the highest PPV (27.7%), and tended to be relatively severe. PPV and OR were not significantly correlated across SE areas (ρ = 0.38, P = 0.14, Spearman). In contrast, several SE areas exhibited ORs that were not significantly greater than 1, and accordingly had PPVs quite close to their low base rates (Fig 4A and 4B). Despite the overall observation that base rate is lower for more severe SEs (Fig 3D), median severity and base rate were not correlated across SE areas (ρ = −0.24, P = 0.36, Spearman), potentially because the SEs within each area were so heterogeneous. Negative predictive values (NPVs) were generally high across all SE areas (range 88–99%; Table X in S1 Table) corresponding to the generally low base rate of any particular SE. All of the findings regarding SE areas were broadly consistent when we removed drugs where an approved indication fell within the same area as the SE (S3 Fig and Table X in S1 Table).

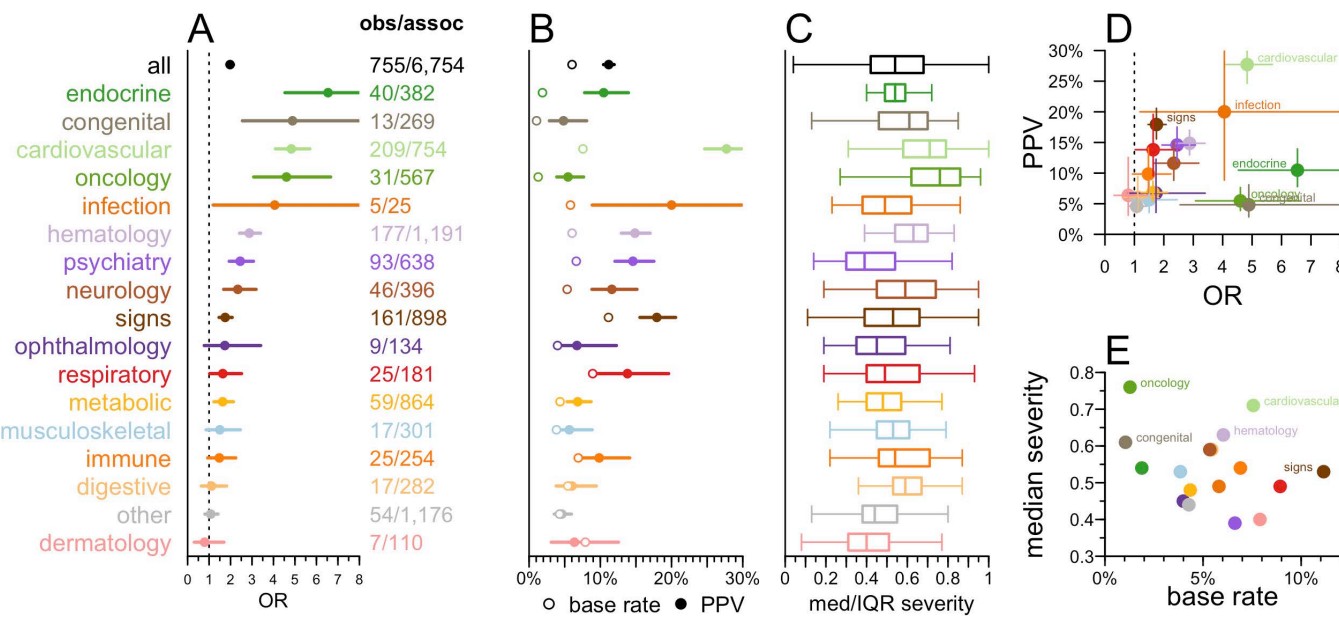

**Fig 4. Utility of human genetic evidence for predicting SEs by affected function or organ system.** A) OR binned by the SE's top-level heading within the Medical Subject Headings (MeSH) ontology. Fractions indicate the number of drug-SE pairs with genetic evidence (denominator) and of those, the number that were observed (numerator). B) Base rate (mean proportion of drugs with the SE) and positive predictive value (proportion of drugs with genetic evidence that exhibit the SE) were binned as in A. C) Median and interquartile range (IQR) of severity quantiles for SEs in each bin. Boxes represent means and interquartile ranges. Whiskers represent minimum and maximum datapoints that fall within 1.5 interquartile ranges of the box. D) Positive predictive value (PPV) vs. OR across and **E)** median severity vs. base rate across SE areas.

The enrichment of genetic evidence among some target-SEs as measured by both OR and PPV were well above the average for any SE area (Tables Y-Z in S1 Table). Three of the five most significantly enriched (ranked by smallest P values) SEs were for cardiovascular traits: hypertension, cardiac arrhythmias, and tachycardia, all of which had ORs > 20 and PPVs > 50% (Table Y in S1 Table). For instance, the seven drug target genes exhibiting genetic evidence to tachycardia (*ACHE, CACNA1C, CHRM2, HCN4, KCNH2, PDE3A, and SCN5A*) serve as targets of a total of 40 drugs, of which 31 drugs, representing six unique targets (all but *HCN4*), have tachycardia reported as a SE, for a PPV of 78% and OR of 35.7 (P = 6.7e-25, Fisher exact test). Hematologic traits including anemia, leukopenia, and thrombocytopenia also ranked near the top of the list, with OR > 8 and PPV > 20% (Table Y in S1 Table). These represent traits that should receive special attention should genetic evidence be found for a discovery-stage target, yielding a high risk of on-target SEs.

## Discussion

Despite encouraging anecdotes and the widespread use of human genetic data to attempt to predict safety risks, only limited systematic evidence has been reported to support the hypothesis that human genetic data can predict drug SEs. Our study provides evidence that SEs are roughly twice as likely to occur for a given drug if that drug's target has a human genetic association to a trait very similar to the SE. This 2.0-fold enrichment was observed even after removing SEs similar to the drug's indication. The magnitude of this enrichment was highly sensitive to the degree of similarity between the SE and the genetically associated trait, which may explain why we observed a larger effect than a previously reported analysis of clinical trial SEs, which grouped only at the level of organ system and observed just a 1.55-fold enrichment after removing SEs similar to the drug's indication.

An important limitation of our analysis is that we do not know which SEs result from on-target pharmacology. By capturing information from drug labels and package inserts, SIDER selects SEs that regulators deemed to be reasonably drug-associated, but this does not guarantee specificity, and coincidental SEs not caused by the drug may be included. More-over, many SEs that are genuinely drug-related will result from off-target and not on-target effects. Our 2.0-fold enrichment is presumably driven solely by the fraction of SEs that result from on-target effects, and this effect would be larger if we were able to filter out off-target and unassociated SEs. We did test for SEs that are enriched among drugs sharing a target, and found a large enrichment of genetic evidence among these, but this test is inherently confounded (see Methods). An additional limitation is that our analysis does not account for direction of effect, either of the drug (agonist vs. antagonist) or of the genetic association. This is primarily due to a lack of annotations in the source databases we used. It is possible that stronger effects would be observed if it were possible to filter for SEs consistent with the drug's mechanism of action and the directionality of gene dosage effect on the associated trait.

Our analysis relied on SIDER for identifying observed drug-SE pairs, because it was the best public database of labeled SEs available, and we were able to readily map most of the SIDER SEs to MeSH terms, a prerequisite for the use of our similarity matrix. Major draw-backs are that SIDER is limited to approved drugs and has not been updated since 2015. Direct queries of the FDA Adverse Event Reporting System (FAERS) would provide an alternative approach, but would present the difficulty of selecting credible associations from raw SE report data. The new database OnSIDES [17], reported after the present work was posted, may provide a more updated extract of drug label information. For either dataset, the task of mapping SEs to MeSH terms remains a formidable one. Like SIDER, both FAERS and OnSIDES are limited to approved drugs. The requirement for drugs to exhibit a favorable risk/benefit balance to achieve approval presumably constrains SEs to be less frequent or less severe than would be the case for drugs in clinical development. It is possible that human genetic evidence has different predictive value for SEs observed in trials that could result in termination than for labeled for approved drugs. A previous study examined SEs in trials [8], but those data are not publicly available due to reliance on the commercial database Cortellis.

Our results demonstrate that human genetic evidence identifies on-target drug mecha-nisms that are at increased risk for SEs among approved drugs. When viewed broadly across all drugs and all SEs, our analysis suggests that genetic evidence has relatively limited positive predictive value (PPV), because the SEs most enriched for genetic evidence are those that are most drug-specific, meaning they have the lowest marginal probability of occurring. Fruitful application of human genetic evidence to SE prediction may benefit from a focus on a subset of SEs that meet all of the following criterial: a) relatively more likely to begin with, b) rela-tively well-predicted by genetics, and c) relatively severe. For example, in our study, genetic evidence was particularly predictive of cardiovascular SEs in general, with a PPV of nearly 30% and relatively high reported severity, while genetic associations to specific traits including tachycardia, arrhythmia, and hypertension were especially predictive. These results support the judicious use of genetic evidence to identify specific SE risks that are worth monitoring and/or mitigating during drug discovery and clinical development.

## Methods

**Side effects.** Side effect (SE) data were obtained from the SIDER database [13] (v4.1), which captures SEs from product labels and package inserts for approved drugs up through 2015. Citeline Pharmaprojects [18] and DrugBank [19] were parsed as described [5,14,20], and SIDER drug names were mapped to Pharmaprojects indications using text matches

to Pharmaprojects drug name synonyms, or, by mapping first to DrugBank using either ATC codes or name matches to obtain CAS numbers, and then looking up CAS numbers in Pharmaprojects; the proportion of drugs mapped by various approaches is provided in Table B in S1 Table. Pharmaprojects matches were used to obtain human gene targets and MeSH terms for approved indications as described [14]. Indications that were Supplementary Concept Records (IDs starting with "C") were mapped to preferred main headings (IDs starting with "D"). SEs were mapped to MeSH terms using UMLS MedDRA – MeSH mapping, exact term and substring match to UMLS and MeSH, and manual curation; the proportion of terms mapped by various approaches is provided in Table C in S1 Table. We removed drugs that were duplicates, lacked an annotated human target, an annotated approved indication, or were unmappable (Table D in S1 Table). Severity rankings were taken from a crowdsourcing study [16]. Ordering of frequency terms was based on numerical values determined empirically with human participants [15].

**Genetic insight.**  As described in Results, one of our approaches to control confounding is to restrict our analysis to SEs that are similar to genetically studied traits. We therefore needed to define which traits count as having been studied genetically, a property we have previously named "genetic insight" [14]. For the purposes of this filter, we considered a trait to have been studied genetically if that trait, or another trait with ≥0.8 similarity, was associated to at least 1 OMIM or IntOGen gene, or was associated to at least 3 GWAS hits at different loci (proxied by chromosome and position rounded to the nearest megabase). For clarity, note that this definition was used exclusively to filter SEs to those that have genetic insight; these criteria are different, and far less stringent, than the criteria used to include genetic associations for the main similarity analysis (see the "Human genetic evidence" section above). A list of side effects lacking genetic insight is provided in Table AA in S1 Table.

**Human genetic evidence.**  We used human genetic evidence from OMIM [21], Open Targets Genetics [22], PICCOLO [23], Genebass [24], and IntoGen [25]; the filtering, aggregation, and MeSH mapping of this dataset has been described [14] but are briefly explained here. The OMIM Gene Map (Sep 21, 2023), which captures Mendelian disease associations, was restricted to solved gene-phenotype associations, removing somatic, drug response, and susceptibility associations, and we also removed associations for which curators determined that credible evidence of causality did not exist. Open Targets Genetics GWAS data (October 12, 2022), which aggregate published GWAS and biobank studies and map them to genes using a machine learning model [22], were restricted to hits with $P < 5e-8$ and gene mappings with ≥50% of the total share of locus-to-gene (L2G) score assigned to any gene. PICCOLO data [23], which map GWAS hits to genes using eQTL colocalization without full summary statistics, were restricted to $P < 5e-8$ and $H4 > 0.9$. Genebass [24] data (October 19, 2023), which capture burden test results from UK Biobank exome sequencing data, were queried using Hail for pLoF (predicted loss-of-function) or missense|LC (missense and low confidence LoF) burdens via SKAT or burden tests with $P < 1e-5$. IntOGen (May 31, 2023) was included to capture enrichment of somatic variants in tumor tissue. Genetic support for the drug's indication (as opposed to for SEs) was taken from a previously published work [9] with a threshold of ≥0.8 similarity between the indication and the genetically supported trait. Summary statistics and examples of drug-indication pairs with genetic support are provided in Tables AB-AE in S1 Table.

**Similarity mapping.**  Similarity was computed as described [14]; the approach is briefly summarized as follows. MeSH terms corresponding to either drug indications (Pharmaprojects), SEs (SIDER), or traits (genetic associations datasets) were included in the matrix. To assign affected function or organ system, the MeSH terms were mapped to their MeSH top level headings as described [14]. A matrix of all possible pairs of resulting

MeSH IDs was constructed. MeSH term Lin and Resnik similarities were computed for each pair as described[34,35]; this approach accounts for both position within the ontological hierarchy as well as term co-occurrence. As before, similarities were set to a minimum of 0, the two scores (Lin and Resnik) were regressed to determine a multiplier used to adjust Resnik scores so that both scores had a range from 0 to 1 with a regression line slope of 1. After this transformation, the two scores were averaged to obtain a single combined similarity score. The similarity matrix is available in the study's online git repository. In the main analyses, we counted as having genetic evidence those drug-SE pairs where the genetic association and SE had similarity ≥0.9 while the genetic association and drug indication had similarity <0.9. Examples for tachycardia are provided in Table AF in S1 Table.

**Target enrichment.** To test whether a particular SE was enriched among drugs with a particular target, we performed a Fisher exact test on the contingency table of drugs with and without the target of interest, with and without the SE of interest reported. Target-SE combinations yielding an odds ratio ≥2 and Benjamini-Hochberg [26] false discovery rate of <0.05 (adjusted from Fisher test P values) were considered to be enriched. It is important to note the limitations and potential confounders inherent in this approach. For example, when a large number of drugs share the same target, there is more statistical power to detect enrichment, while when there is but a single drug for a target, it is impossible to assess enrichment. Moreover, drugs sharing a target may also be of the same chemical class and may therefore share off-target liabilities. In consideration of these limitations, we considered target enrichment among our candidate variables in Fig 1A but did not use this metric in ensuing analyses.

**Models and statistics.** Analyses used custom scripts in R 4.2.0. The primary metric in this analysis — whether an SE is more likely to be observed when there was genetic evidence — was computed as an odds ratio (OR) from a Fisher exact test on the 2 x 2 contingency table of drugs with and without an SE, whose targets do or do not have a genetic evidence. Following the findings of Fig 1A and 1B, this was computed after removing drugs with an indication similar to the SE, and after removing SEs not studied genetically. The shaded areas for curves and error bars in forest plots represent the 95% confidence intervals from this Fisher test. Binomial logit models used the SE's occurrence as the dependent variable, and genetic evidence and the variable of interest (for instance, SE severity) as independent variables, with interaction terms — in R, glm(observed ~ sim_assoc * severity, family='binomial'). The Cochran-Mantel-Haenszel (CMH) test for heterogeneity was performed across these 2 x 2 contingency tables for each MeSH area. For analysis of attributes that are only defined when the SE is observed (frequency and placebo status in Fig 2), the Fisher test was based on the 2 x 2 contingency table of drugs that do or do not have an SE *with the stated attributes* (for instance, frequency >10% in patients), whose targets do or do not have genetic evidence. In such instances, binomial logit models used presence of genetic evidence as the dependent variable, and the variable of interest (for instance, numerical frequency) as the independent variable: e.g., glm(sim_assoc ~ frequency, family='binomial'). Frequency terms were treated alternatively as ordinal variables, resulting in terms for linear and higher-order terms, or as numerical variables with the term's rank as the numerical value, resulting in only a linear term. Base rate, or an SE's drug specificity, was defined as the proportion of drug-SE pairs for which the SE was observed. Positive predictive value (PPV) was defined as the number of drug-SE pairs where the SE was observed and was supported by genetic evidence, divided by the total number of drug-SE pairs where the target had genetic evidence. Linear regression for specificity versus severity used the logarithm of the number of drugs for which an SE was observed, in R: lm(log(n_drugs) ~ severity). Correlations across MeSH areas were tested

using Spearman rank correlations. All tests were two-sided, and P values less than 0.05 were considered to be nominally significant.

## Supporting information

**S1 Fig. Examination of possible confounders and establishment of metric used throughout.** A) Correlogram showing the odds ratios (ORs) by Fisher exact test for enrichment of all combinations of properties (S1 Table and Methods) evaluated in the dataset. **B)** OR for enrichment of genetic evidence vs. SE observed, with the indicated filters applied.
(TIFF)

**S2 Fig. Breakdown of evidence sources for oncology.** A) Forest plot of OR by source of evidence (IntOGen somatic evidence vs. all sources of germline evidence) versus oncological and non-oncological SEs. **B)** Drug specificity of oncological and non-oncological SEs. IntOGen overall has an OR < 1 because its somatic evidence are almost exclusively similar to oncological SEs, which are more drug-specific than non-oncological SEs. Thus, the IntoGen OR for oncology only is shown in Fig 1. Germline evidence appears to have a higher OR than somatic evidence for oncology. Note that the germline evidence for oncology is driven by GWAS associations for X genes: CYP19A1 (endometrial neoplasms), ESR1 (breast neoplasms), FGFR2 (neoplasms), INSR (polycystic ovary syndrome), and SRD5A2 (breast neoplasms).
(TIFF)

**S3 Fig. Breakdown by side effect area.** As Fig 4, but within each MeSH area, any drug with any indication in that area is removed.
(TIFF)

**S1 Table. Table A.** Properties of drug-side effect matrix. **Table B**. Methods of matching SIDER drug names to Pharmaprojects. **Table C**. Methods of matching SIDER side effect names to MeSH terms. **Table D**. Reasons SIDER drugs drop out of analysis. **Table E**. Cross-tabulation of drug-SE properties. Table F. Effects of requiring genetic insight and removing similar indications. **Table G**. Sensitivity to similarity threshold for inclusion of similar genetic associations. **Table H**. Sensitivity to similarity threshold for removal of similar indications. **Table I**. Breakdown by source of genetic evidence. **Table J**. Breakdown by somatic vs. germline and oncology vs. non-oncology. **Table K**. Properties of oncology vs. non-oncology SEs **Table L**. Binned analysis of numerical SE frequency. **Table M**. Logit model coefficients for numerical SE frequency. **Table N**. Binned analysis of SE frequency terms. **Table O**. Logit model coefficients for SE frequency terms, ordinal model. **Table P**. Logit model coefficients for SE frequency terms, linear term only. **Table Q**. Binned analysis of placebo status. **Table R**. Logit model coefficients for placebo status. **Table S**. Binned analysis of SEs by drug specificity (number of drugs where the SE is observed). **Table T**. Binned analysis of SEs by severity quartile. **Table U**. Logit model coefficients for severity analysis. **Table V**. SE drug specificity versus severity bin. **Table W**. Linear model coefficients for SE severity vs. drug specificity. **Table X**. Breakdown by MeSH area. **Table Y**. Enrichment statistics by GWAS association MeSH term. **Table Z**. Enrichment statistics by side effect MeSH term. **Table AA**. Side effects lacking genetic insight, by number of drugs **Table AB**. Count of drug-indication pairs with and without genetic support. **Table AC**. Drug-indication pairs with genetic support. **Table AD**. Count and base rate of drug-SE pairs by genetic support status. **Table AE**. OR by genetic support status, with and without sim_indic filter. **Table AF**. Details of drugs whose targets are genetically associated to tachycardia.
(XLSX)

## Author contributions

**Conceptualization:** Matthew R Nelson.

**Data curation:** Eric Vallabh Minikel.

**Formal analysis:** Eric Vallabh Minikel, Matthew R Nelson.

**Investigation:** Eric Vallabh Minikel, Matthew R Nelson.

**Methodology:** Eric Vallabh Minikel, Matthew R Nelson.

**Supervision:** Matthew R Nelson.

**Visualization:** Eric Vallabh Minikel.

**Writing – original draft:** Eric Vallabh Minikel.

**Writing – review & editing:** Eric Vallabh Minikel, Matthew R Nelson.

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
