## [Decision Letter · Decision Letter 0]

15 Oct 2024

Dear Dr Nelson,

Thank you very much for submitting your Research Article entitled 'Human genetic evidence enriched for side effects of approved drugs' to PLOS Genetics.

The manuscript was fully evaluated at the editorial level and by independent peer reviewers. The reviewers appreciated the attention to an important topic but identified some concerns that we ask you address in a revised manuscript.

We therefore ask you to modify the manuscript according to the review recommendations. Your revisions should address the specific points made by each reviewer.

2) Upload a Striking Image with a corresponding caption to accompany your manuscript if one is available (either a new image or an existing one from within your manuscript). If this image is judged to be suitable, it may be featured on our website. Images should ideally be high resolution, eye-catching, single panel square images. For examples, please browse our archive . If your image is from someone other than yourself, please ensure that the artist has read and agreed to the terms and conditions of the Creative Commons Attribution License. Note: we cannot publish copyrighted images.

If present, accompanying reviewer attachments should be included with this email; please notify the journal office if any appear to be missing. They will also be available for download from the link below. You can use this link to log into the system when you are ready to submit a revised version, having first consulted our Submission Checklist .

PLOS has incorporated Similarity Check , powered by iThenticate, into its journal-wide submission system in order to screen submitted content for originality before publication. Each PLOS journal undertakes screening on a proportion of submitted articles. You will be contacted if needed following the screening process.

To resubmit, log into your Editorial Manager account and select the option 'Revise Submission' in the 'Submissions Needing Revision' folder.

Yours sincerely,

Lynn Petukhova, Ph.D.

Guest Editor

PLOS Genetics

Hua Tang

Section Editor

PLOS Genetics

Reviewer's Responses to Questions

**Comments to the Authors:**

Reviewer #1: Described in this manuscript is a detailed analysis of how human genetics can help predict the occurrence of drug side-effects (SEs). To that effect, the Authors used extensive databases on marketed drugs and their reported SEs, matching them with databases on the genetic basis of traits similar to the SEs. They observed that the risk of SE in presence of a genetic association between the drug target gene and latter trait was increased by 2-fold. From this observation, they suggest that incorporating human genetics could help predict future SEs for investigational drugs.

The topic is very timely and relevant, not only for the pharma industry (SEs are responsible for a substantial proportion of failures in development), but also for public health, considering that SEs are responsible for 5-10% of hospital admissions.

The work presented here is a logical extension of previous, outstanding work performed by the same Authors on the value of human genetics to predict the probability of success for a drug target to eventually go through the hurdles of drug development to regulatory approval.

The manuscript is well written. Methods are sound and sophisticated, yet well presented. Data is well presented and carefully interpreted, conclusions are solidly supported by the data and potential implications of the findings cautiously presented.

In my opinion, the text is quite dry, and the manuscript would greatly benefit from specific examples that shall illustrate the concept, and the type of analyses performed. In particular, Readers would appreciate to see the name of a drug, the type of ADR, traits similar to SEs, examples of SEs which are very rare to common, etc…

The description of PPVs is highly relevant, although, as discussed by the Authors, they pertain here to approved drugs and may not necessarily apply to drugs in development. In my opinion, it may be worthwhile to emphasize the fact that PPVs are hardly better than base rate (i.e. not informative) for SEs in specific functions/organs.

PPVs clearly stand out for cardiovascular SEs. This is somehow surprising considering that PPVs are much lower for metabolic traits. A deep dive into this specific type of SEs would be appreciated.

A discussion on NPV would be appreciated, i.e. how predictive for SEs is the absence of any genetic association ? Maybe this cannot be analyzed.

The absence of correlation between base rate and severity is intuitively surprising, considering the fact that certain drugs would be approved with severe SEs, as long as these SEs are very rare. Any explanation to Figure 4E ?

Reviewer #2: Minikel & Nelson focus a different lens on the data & methods assembled for their recent publication (Minikel et al. Nature, 2024; Minikel et al. medRxiv, 2023), to assess the genotype:phenotype correlation of side effects as opposed to lead indication success in clinical trials. One thing that the authors don't discuss but I believe is important, is what was their null hypothesis here? Given their previous publications and that they filtered their data to enrich their analysis with side effect traits that have been genetically investigated, null was that you'd have positive OR. And in fact, in aggregate that's what they see with and OR ~2 of what Nelson published in 2015 for association of genetics with drug approval. Pharmacological perturbation recapitulating genetic associations should have been the null in my opinion, therefore the results follow the null and the results seen in previous investigations (ref. 7 & 8). Was the reason for this work to more fully test the results of ref. 8?

The potential impact of the results of this investigation is limited because the side effects investigated are of approved drugs, therefore their liability to a development program are minimal and would be unlikely to influence go/no-go decisions for programs. The results presented are also very high-level and very heterogeneous (Fig 4), therefore also difficult to draw any strong conclusions nor think about how to operationalize this information into any practice in drug development. To no fault of the authors, the data available to analyze for the study is almost 10 years old and therefore limits an ability to see if the trends identified here have changed at all since the publication of Nelson et al. Nature Genetics, 2015 which has influenced the number of targets entering drug development with human genetic evidence.

Perhaps beyond the scope the authors intended for this work, but to me the most interesting (and potentially useful) questions have not been asked and important aspects of the data not discussed. I'll detail these critiques here:

1. I'd be interested to know the distribution of the primary indications for the approved drugs analyzed here, the distribution of year of approval, and the median number of SEs. Is there a skew in indications that might have a skewed risk:benefit ratio such that some indications can get approved with more and/or more severe side effects? Has the number of SEs for drugs changed over time of approval, for example do newer drugs have fewer SEs and older drugs? Does this information confound the results of these results at all?

2. Is the OR of the genetic association to SEs at all associated/correlated with whether there is genetic evidence for the target in the primary indication?

3. Are the targets that are associated with SEs more pleiotropic (i.e. have more associations to different phenotypes) than the "average" (could be defined in many ways) gene?

4. Is the direction of effect between the pharmacological perturbation and the genetic signal consistent or inconsistent?

a) This is the biggest question that pops out to me from the manuscript. You state that you've removed SEs that are similar to the approved indication. How frequent was this? This is a very surprising result because it suggests that a single target has completely different effect directions on very similar traits.

b) Extension of question 3 but in terms of directionality. If the target for the approved indication is a pleiotropic target, then you'd expect the gene has to have different directions of effect on the approved indication and the SE phenotype. Is this in fact true on average or not?

5. Was there a difference between the modality of the approved drug (small molecule vs. antibody vs. etc.)? Potential proxy for the specificity of the approved molecule.

a) perhaps related question about specificity, what are the gene classes (kinases, receptors) of the targets whose SEs have genetic evidence. Are there close paralogs in the genome or highly similar genes with similar genetic associations?

b) For the analysis in Figure 3, you ignored whether the SE was from targeting the same gene. This seems like a missed opportunity to game more insight. If not the same gene, but part of the same protein complex or known protein-protein interaction that provides something that could be operationalized in drug development just like for efficacy determination.

6. Perhaps I just don't understand Figure 1, as the language used to describe what is being shown is a bit hard to follow. But I believe that Fig1A is showing the OR of genetic evidence for the reported SE as you vary the phenotypic similarity between the stated SE and the proxy phenotype you use to look for genetic evidence. Given this understanding this figure makes absolutely no intuitive sense to me. The first part of the curve is the expectation, the further the SE is from the "genetic trait" the lower the OR is and essentially is an OR ~1 at around a similarity of ~ 0.7. But then the curve increase as you approach similarity of 0.5 and hits a new local minimum at a similarity of 0.2. There is no exploration or discussion of this fact in the manuscript. Is this simply a power thing given the number of observations detailed on the x-axis. Or this this noise? How can you account for this?

a) Generally, I don't have a strong intuition for the trait similarity metric and how it behaves. Just because it is a value between 0 and 1 doesn't mean that its behavior is similar across the spectrum. How much of the results are because the performance of the metric? Is the difference between 0.2 and 0.5 the same between 0.5 and 0.8? How have you chosen thresholds here and how does that relate to attributes of the metric? How different would results be if you used HPO or some other phenotype ontology instead of MeSH?

7. Fig2 also a bit confusing. What is different about the "common" and "frequent" class? The fact that "frequent" has an OR ~1 while "common" has an OR ~3 makes no sense, especially given that the numerical frequency plot above shows a significant jump in OR as you get above 10% frequency.

8. Wouldn't you expect that the base rate of severe SEs (Fig3D) to be the lowest given that those SEs would make the risk:benefit ratio get to the point where regulators wouldn't approve the drug? What primary indications had the highest frequency of severe SEs?

9. Perhaps I also don't understand Figure 4 because of the language used to describe what is being shown. But the description, "binned by the SE’s top-level heading within the Medical Subject Headings (MeSH) ontology.", says to me that the terms on the y-axis of the forest plot are the SEs. Which would mean, that there are approved drugs where the SE is cancer?!? What does it mean to have an SE that is congenital? Does this mean that if pregnant women are given the drug their babies have issues?

a) What is the association between the indication that the drug is approved for and having SEs in endocrine/congenital/cardiovascular/oncology/infection?

b) I also don't understand this statement, "Fractions indicate the number of drug-SE pairs with genetic evidence (denominator) and of those, the number that were observed (numerator)." If a drug-SE pair has genetic evidence then by definition wouldn't it have to have been observed? Can you explain in more detail what the numerator and denominator are counting because I don't get it.

10. In Fig1C are you surprised that OMIM/Genebass evidence provides weaker evidence than GWAS and oncology evidence? This is the reverse of the indication approval evidence. Is this evidence that the drugs aren't hitting the target hard enough or is this somehow related to trait similarity between the SE and the OMIM phenotype?

a) are you surprised that germline oncology evidence is so enriched? What genes are represented here? Is there some confounding here?

11. Small omissions and grammar changes:

a) First sentence of introduction: Put a metric here, don't make the reader go to the reference to figure it out. If it is major, how major is it?

b) 4 sentence of the 2nd paragraph of the introduction: "Off-target SEs occur AS the result..." NOT "Off-target SEs occur are the result..."

c) 2nd sentence of the 2nd paragraph of the results: I believe a p-value is missing in the parentheses in front of "Fisher's exact test".

The manuscript can be published as is, after correcting minor formatting issues highlight in point 11 and addressing some the ambiguity in definitions in figures, but I believe the impact and insight provided by it will be greatly reduced than if the authors perform the additional analyses suggested. Currently the manuscript is an expansion and replication of previous work and therefore inline with expectation, and leaves the drug developer unsure what to do with the results; which perhaps is fine, just that genetics and pharmacological perturbations of genes provide similar results.

**Have all data underlying the figures and results presented in the manuscript been provided?**

Reviewer #1: Yes

Reviewer #2: Yes

PLOS authors have the option to publish the peer review history of their article (what does this mean? ). If published, this will include your full peer review and any attached files.

**Do you want your identity to be public for this peer review?** For information about this choice, including consent withdrawal, please see our Privacy Policy .

Reviewer #1: No

Reviewer #2: **Yes: ** Aaron G. Day-Williams

---

## [Decision Letter · Decision Letter 1]

26 Feb 2025

Dear Dr Nelson,

We are pleased to inform you that your manuscript entitled "Human genetic evidence enriched for side effects of approved drugs" has been editorially accepted for publication in PLOS Genetics. Congratulations!

Yours sincerely,

Lynn Petukhova, Ph.D.

Guest Editor

PLOS Genetics

Hua Tang

Section Editor

PLOS Genetics

Aimée Dudley

Editor-in-Chief

PLOS Genetics

Anne Goriely

Editor-in-Chief

PLOS Genetics

Comments from the reviewers (if applicable):

Reviewer's Responses to Questions

**Comments to the Authors:**

Reviewer #1: The Authors have adequately responded to my comments.

I appreciate that they have inserted a few specific examples to illustrate the concepts presented here. One has captured my attention which, in my opinion, exemplifies the strengths and limitations of the analyses presented here, including the limited predictive value of genetic evidence. This example is topiramate "which is indicated for migraine, targets SCN5A and has cardiac arrhythmia listed as a SE" (Results section, paragraph 2).

In the clinic, topiramate is mostly used as anticonvulsant and for migraine prevention. From a pharmacology perspective, its mechanism of action is poorly understood, and appears to involve inhibition of sodium and calcium channels, activation of GABA receptors and other mechanisms like inhibition of carbonic anhydrase. The drug has numerous adverse effects, some of them being severe, including glaucoma, hyperthermia, kidney stone, metabolic acidosis, suicide and depression (which are all traits for which extensive GWAS have been published), whereas cardiac AEs are very rare, mostly bradycardia. In fact, association between topiramate and Brugada syndrome (which is genetically associated with SCN5A) has only been described in case reports.

In my opinion, the manuscript would greatly benefit if the Authors would dig deep into this particular example (for instance, are any of the genes encoding topiramate targets associated with the traits listed above ?) and use it to critically describe the strengths and limits of their paper.

Reviewer #2: The authors have thoroughly addressed all of my requests for clarifying language and analyses that have greatly improved the manuscript. It is a great body of work and ready for publication

**Have all data underlying the figures and results presented in the manuscript been provided?**

Reviewer #1: Yes

Reviewer #2: Yes

PLOS authors have the option to publish the peer review history of their article (what does this mean? ). If published, this will include your full peer review and any attached files.

**Do you want your identity to be public for this peer review?** For information about this choice, including consent withdrawal, please see our Privacy Policy .

Reviewer #1: No

Reviewer #2: **Yes: ** Aaron G. Day-Williams

**Data Deposition**

http://datadryad.org/submit?journalID=pgenetics&manu=PGENETICS-D-24-00658R1

**Press Queries**

---

## [Editor Report · Acceptance letter]

PGENETICS-D-24-00658R1

Human genetic evidence enriched for side effects of approved drugs

Dear Dr Nelson,

We are pleased to inform you that your manuscript entitled "Human genetic evidence enriched for side effects of approved drugs" has been formally accepted for publication in PLOS Genetics! Your manuscript is now with our production department and you will be notified of the publication date in due course.

With kind regards,

Anita Estes

PLOS Genetics

On behalf of:
